# Change in Pull-Out Force during Resorption of Magnesium Compression Screws for Osteosynthesis of Mandibular Condylar Fractures

**DOI:** 10.3390/ma14020237

**Published:** 2021-01-06

**Authors:** Marcin Kozakiewicz

**Affiliations:** Department of Maxillofacial Surgery, Medical University of Lodz, 113th S. Żeromskiego, 90-549 Lodz, Poland; marcin.kozakiewicz@umed.lodz.pl

**Keywords:** screw, pull-out force, magnesium, polylactide, polyglycolic acid, fixation material, mandible, condylar head fracture

## Abstract

Background: Magnesium has been used as degradable fixation material for osteosynthesis, but it seems that mechanical strength is still a current issue in these fixations. The aim of this study was to evaluate the axial pull-out force of compression headless screws made of magnesium alloy during their resorption. Methods: The tests included screws made for osteosynthesis of the mandible head: 2.2 mm diameter magnesium alloy MgYREZr (42 screws) and 2.5 mm diameter polylactic-co-glycolic acid (PLGA) (42 pieces, control). The screws were resorbed in Sørensen’s buffer for 2, 4, 8, 12, and 16 weeks, and force was measured as the screw was pulled out from the polyurethane block. Results: The force needed to pull the screw out was significantly higher for MgYREZr screws than for PLGA ones (*p* < 0.01). Within eight weeks, the pull-out force for MgYREZr significantly decreased to one third of its initial value (*p* < 0.01). The dynamics of this decrease were greater than those of the pull-out force for PLGA screws (*p* < 0.05). After these eight weeks, the values for metal and polymer screws equalized. It seems that the described reduction of force requires taking into account when using magnesium screws. This will provide more stable resorbable metallic osteosynthesis.

## 1. Introduction

The basic treatment method for dislocated fractures is osteosynthesis with metal alloy materials [1,2,3,4,5,6]. Materials, especially titanium alloys, have appropriate physical properties [7,8], but in terms of biochemical and long-term impacts on the human body, they raise serious concerns, i.e., inflammation, thermal hypersensitivity, cell apoptosis, and oxidative and nitrosative stress in treated patients [9,10,11,12]. If the alloy comes into contact with the oral cavity, its relatively rapid surface degradation [13] causes undesirable biological reactions of surrounding soft and hard tissues, loss of osseointegration weakening the maintenance of the implant in the bone, chemical reactions, functional stresses, and bacterial attack [14]. For this reason, and because of the mechanical irritation of surrounding tissues during movement in the joints [15], a planned removal of the entire fixation material 2–3 months after implantation is postulated [15,16]. The solution to these problems would be to use a fixing material that would disappear after the period of bone healing. Therefore, interest has been directed toward biodegradable materials [17,18].

Conventional osteosynthesis may be replaced by bone adhesive in the next decades. Such attempts have been made for many years, and research on bone adhesive materials is still ongoing [19]. However, there is currently no material available that combines all of the important clinical features. The main cause of this condition is difficulties in the production and adaptation of bone adhesives. This is attributable to the challenging conditions in which bone adhesives are used. Current adhesives are not able to combine three requirements: biocompatibility, degradability, and bond strength. Genetic and tissue engineering, as well as biotechnology, will increase the effectiveness of producing an adhesive for bonding mandibular head bone fragments in the future [19,20].

Currently, two other groups of fixing materials are available for resorbable osteosynthesis in mandibular condylar fractures: polymers and metal alloys. Because of the promising chemical properties [20], a series of polylactide fixing materials [21,22] dedicated for that purpose (pins, screws, and meshes) have been developed in the last two decades. Their physical properties require some compromise in bone fragment fixation. These materials have limited clinical application in load-bearing fractures, and such a fracture is a fracture of the mandible condyle. It was reported that the largest gap distance in the fracture site was observed for poly-L-lactide (PLLA), followed by magnesium alloy fixation. The width of the fracture gap between bones increased with respect to increased masticatory loading in both materials [23]. To counter these challenges, poly(lactic-coglycolic) acid (PLGA) implants aim to combine favorable resorption time and to reduce local reactions while maintaining sufficient stability to allow bone healing [24,25,26]. Satisfactory results have been presented using PLGA fixation without resorption-related complications [27,28,29], and it seems to be a proper material for maxillofacial surgery [30].

In recent years, magnesium alloys have been observed to have better mechanical properties than polymers used for osteosynthesis and still maintain the advantage of resorbability [31,32] and have also been used in bone regeneration for their biological proprieties [33]. However, it is known that the mechanical properties of magnesium implants are lower than those of medical titanium [23]. That study [23] was based on a numerical model. Therefore, further studies should investigate resorbable metal alloys. It seems that mechanical strength is still a current issue in magnesium fixation. This problem is worth addressing because maxillofacial surgeons and orthopedists are currently intuitively sure of their osteosynthesis strength, but these fixations are mainly made with titanium alloys [34,35,36,37,38]. The only group of medical specialists who are fully ready to use magnesium screws are children’s traumatologists, who have their own experience in using PLLA and PLGA materials [39,40,41].

When considering metal resorbable bone osteosynthesis, it is worth asking how stable early healing results can be expected to last (please see the figure presented in the Discussion section). There may be several reasons for the condition presented in that figure, but among these reasons is the weakening of the force keeping bone fragments stationary by the magnesium screws used. Bone fixation keeps the bone fragment in a reduced position until bone continuity is restored. This period of bonding takes quite a long time. Full remodeling of the new bone at the fracture site takes up to one year after the fracture [42]. The first adhesion occurs in six weeks, but the bone at the healing site does not yet have its original physical characteristics. Therefore, there is a need to maintain the internal immobilization of bone fragments for several months. This process is to be ensured by the screws, and their physical characteristics should allow the fragments of the bone to remain immobile relative to each other. If the material is resorbable, it seems possible that its ability to hold the bone in place may change over time. The lack of maintenance of bone fragments in place causes bone deformation and functional disorders in the patient.

The aim of this study was to evaluate the axial pull-out force of compression headless screws made of magnesium alloy during their resorption.

## 2. Materials and Methods

The research presented here is a continuation of experiments published in 2020 [43]. The same method and the same measuring station were used for axial pull-out tests after the resorption time was added.

The tests included 84 screws of 14 mm length made for osteosynthesis of the mandible head by ChM (www.chm.eu; Juchnowiec Koscielny, Poland): 2.2 mm diameter magnesium alloy MgYREZr, i.e., WE43 MEO 42 screws and 2.5 mm diameter poly(lactic-coglycolic) acid in a molar ratio of 85:15 (42 pieces). Polylactic-coglycolic acid (PLGA), which is considered a good alternative to titanium in osteosynthesis, was chosen as the control [24,25,26,27,28,29,30].

Polyurethane foam blocks were used in this study. The high variability in the density and elastic modulus of the bone affects the results of biomechanical tests [34]. Compared with human bone, synthetic foam materials have been shown to yield less intra- and interspecimen variability (www.astm.org/Standards/F1839.htm). These blocks have consistent material properties that are similar to those of human bone. Solid polyurethane foam is widely used to mimic and is an ideal medium for mimicking human bone, and the American Society for Testing and Materials [35,36] has established it to be a standard material for testing orthopedic devices and instruments. In this study, polyurethane foam with a density of 0.64 g/cm^3^ (Sawbones Europe AB, Krossverksgatan 3, 216 16 Malmö, Sweden) was used as a substitute for bone [37,44,45]. The test methods used the followed standard F543 (www.astm.org/Standards/F543.htm) for medical bone screws. The axial pull-out strength was used in this study to compare two types of mandible head fixation screws. The MTS Insight 100 kN testing system with the force detected with the Interface 1010ACK-1.25KNB model was 1.25 kN, and the displacement detected was ±50 mm (MTS Insight 100, MTS Systems 14000 Technology Drive, Eden Prairie, MN, USA) and was used to determine the axial pull-out strength of the screws. TestWorks 4 (MTS Systems 14000 Technology Drive, Eden Prairie, MN, USA) was used as software. The test velocity used was 5 mm/min. at a temperature of 23 ± 2 °C. The screws were tested at an insertion depth of 6 mm into a polyurethane block.

The screws screwed into test blocks were placed in Sørensen’s phosphate buffer consisting of monobasic kalium phosphate (KH_2_PO_4_) and dibasic sodium phosphate (Na_2_HPO_4_) in a volume ratio of 18.2% and 81.8, which stabilized the pH of the buffer at 7.4 ± 0.2. The experimental time intervals were 0, 2, 4, 8, 12, and 16 weeks (Figure 1). After each of these periods, seven screws with blocks were removed from the buffer and tested according to the above protocol.

Statistical analysis was performed in Statgraphics Centurion 18 (Statgraphics Technologies Inc., The Plains, VA, USA). A *T*-test was applied to compare the means (normally distributed variables), and a Mann–Whitney W-test was used to compare the medians (non-normally distributed variables) of the two independent samples for axial pull-out force comparison at the same time moment between MgYREZr versus PLGA screws. The same tests were used to assess the decrease in pull-out force with the duration of resorption.

## 3. Results

As the experiment progressed, the magnesium alloy screws were covered with a white coating, and pits could be observed on their surface. The poly(lactic-coglycolic) acid screws material became less transparent and white (Figure 1). The force needed to pull the screw out was significantly higher for magnesium screws than for polymer screws (*p* < 0.01). All data are presented in Table 1 and Figure 2. The evaluation of changes depending on the time of the experiment revealed that in the case of PLGA screws, the loss of pull-out force was significant in relation to the initial force only after 8 (*p* < 0.05), 12 (*p* < 0.01), and 16 weeks (*p* < 0.01). For magnesium screws, however, a decrease in pull-out force was noted in all test periods (*p* < 0.01).

Within eight weeks, the pull-out force for magnesium alloy screws was significantly reduced to one-third of its initial value. The dynamics of this decrease were greater than those of the pull-out force for polymer screws. After these eight weeks, the values for metallic and polymer screws equalized.

## 4. Discussion

Currently, osteosynthesis is based on screw connections. The role of metallic resorbable screws is to heal the bone, and then the fixing material disappears from the environment. For this reason, in recent years, this class of resorbable fixation materials has aroused considerable interest in maxillofacial surgery [40,46]. Other advantages include anti-inflammatory [47,48] and antibacterial activity [49,50,51] as well as bone formation induction [52,53]. However, primacy leads to the fact that this fixation material will not have to be removed from the body through a second surgical procedure. These advantages have been recognized in both orthopedics [46] and maxillo-facial surgery [54,55].

Nowadays, the magnesium alloys used have twice as low Young’s modulus (E = 45,000 MPa) as commonly used titanium grade 5 and 23 (96,000 MPa); approximation of this previous value to the condition of compact bone may be considered a biological advantage of magnesium fixation. Despite this lower E value, the modulus of elasticity in the tension of magnesium alloy is still much higher than that of clinically used resorbable polymers (3000–10,000 MPa) [23]. These conditions have an obvious influence on the lower stability of PLGA fixation, and it is worth considering that they may affect the stability of the screw pulling out of its original position caused by screw absorption (Figure 3).

The pull-out force quickly decreases in magnesium screws. The rate of this decrease was lower for polymer screws, but it should be noted that the initial pull-out force was also significantly lower for PLGA screws than for magnesium screws (*p* < 0.01). This effect has already been observed previously [43]. It is necessary to comment on the observation of increased pull-out force in the twelfth week of the magnesium alloy experiment. This is difficult. The mechanism of magnesium resorption in an aqueous environment is not fully known [56]. Perhaps it is related to the formation of crumbs, scales, and particles of the alloy on the surface as a result of the progressing absorption of the screw. Such defragmentation of the MgYREZr surface may increase the friction during the pulling-out of the screw, which affects the higher pull-out force.

First, the screw diameter certainly has an impact on the pull-out force [54,57]. The larger the diameter, the higher the force. Second, thread depth, taper shape, and taper length have an effect on the pull-out force [58]. Increasing the “taper depth” reduces the pull-out force (probably as a result of just decreasing the diameter of the screw core as a result of increasing the thread depth). The increasing angle of the “taper” in the screw reduces the pull-out force. Moreover, increasing the taper length, i.e., the number of threads on the tapered core, increases the pull-out force of the screw. However, this applies to screws made of the same material. In this experiment, the implant material had the greatest influence on the result. This result was due to the significant difference in Young’s module of MgYREZr and PLGA.

The decomposition of PLGA implants leads to the formation of single particles of glycolic and lactic acids. Glycolic acid is converted into glyoxylate and then under the influence of glycine transamidase to glycine. Some glycolic acid monomers are excreted in urine. After conversion to pyruvic acid, glycine is processed into carbon dioxide and water. In turn, the lactic acid molecules are more directly converted to pyruvic acid and further metabolized to carbon dioxide and water. [59]. Resorption of magnesium occurs when the alloy is exposed to an aqueous environment inside the human body. Magnesium hydroxide and hydrogen gas are produced as a result of magnesium resorption. Mg(OH)_2_ reacts with chlorine ions in the body’s internal environment, creating easily soluble MgCl_2_. This process leads to the resorption of the implant with the formation of gaseous hydrogen and ionized hydroxyl groups [46]. It seems that with the accumulation of hydrogen around the relatively small volumes of magnesium alloy, the screws dedicated to the fixation of the mandible head [43] would be less of a clinical problem with particles and molecules persisting in the surrounding soft tissue after degradation of the main mass of the PLGA screw [18,25].

The time of mandibular bone fracture healing, including mandibular head healing, is six weeks. After that, remodeling lasting several weeks follows. During this time, it would be good for the fixing material to keep the bone fragments unchanged despite the masticatory forces. In numerous comparative studies, it was noted that the fracture gap after osteosynthesis with resorbable material increases depending on the force with which it bites. A 200 N masticatory loading force opens the gap to 0.111 mm for magnesium and 0.299 mm for polymer. For 600 N, it was 0.318 mm for magnesium and 0.848 mm for polymer [22]. These narrower gaps will be less damaging to the fracture healing process, but the author recommends some caution even in the application of the magnesium screws.

Resorbable fixing material is used to achieve many goals. Polymeric fixations are clinically satisfactory for use in children and orthognatic surgery to avoid resurgery for their removal. However, their limitations should be considered: physical strength, troublesome plate shaping, and taping of the screw canal or transparency for X-rays, making it impossible to assess their position after surgery. Magnesium alloys have remained of great interest in recent years because of their strength, biocompatibility, radiopacity, and resorbability, and they need to be verified for widespread use in humans [31]. As an example, the pull-out force during magnesium screw resorption exceeding the value of PLGA can be cited here. In the available literature, there were no results of tests describing thermal hypersensitivity, cell apoptosis, and oxidative and nitrosative stress in the case of using screws made of magnesium alloys. The only available information concerns the evaluation of inflammation. No higher frequency of inflammatory reactions was observed in patients after magnesium osteosynthesis [59].

To date, there is no ideal material for osteosynthesis. On the one hand, in the case of polymers, edema persisting for several months around the foreign body is described [25,60]. On the other hand, in the case of magnesium alloys, hydrogen production around the fixation for a few months is known (Figure 3E). It now seems that the use of resorbable metal alloys with a higher stiffness than resorbable polymers for mandibular head fixation [61] is a good option for traumatology patients provided that surgeons know the mechanical properties of the material, which influences changes in the forces holding the screws in place during bone healing together with metal resorption (Table 1).

There are some limitations in this experiment. The study did not examine the influence of osseointegration on the screw pull-out force change. The osseointegration of the bone to the screw will probably reduce the purely mechanical effect of that force loss during resorption. On the one hand, it is worth emphasizing here that some impact is to be expected in this respect because in the animal model, a low implant resorption rate was determined in the first 16 weeks [62]. On the other hand, it should also be expected that the mechanical significance of osseointegration is slightly less in magnesium alloys than in titanium alloys [63]. In this light, it seems that the mechanical properties of the material used for screws and the limitation of the patient’s mastication force determine the maintenance of the position of bone fragments after fixation during healing.

After this study, several clinical remarks need to be made. The use of several screws for mandibular head osteosynthesis is more reasonable than the use of only a single screw because magnesium alloy quickly loses its initial mechanical properties with the time of resorption. This difference is distinct from the use of standard low profile screws made of titanium alloys [57,64]. This issue is reduced and slightly supported by a modification that raises the diameter of magnesium screws to 2.2 mm [43], which improves the mechanical properties of the screw. Nothing stands against the application of a thicker screw with additional 2–3 narrower screws (1.7 mm). It is still suggested that the screwdriver socket should be handled gently because of the fragile construction of the cruciform design [43].

The use of implants based on magnesium alloys as resorbable metals for osteosynthesis shows effective possibilities for their utility in osseosurgery. There are no differences between magnesium and titanium fixation materials with regard to biocompatibility and frequency of complications. Therefore, magnesium materials should be considered for clinical applications in maxillofacial surgery and orthopedics [65], and they are much more suitable than polylactoglycolic materials [66]. More randomized controlled trials or prospective studies are needed to demonstrate that different resorbable materials are better or comparable to titanium screws [67,68]. With regard to the future development of the use of magnesium alloys, further research is needed to increase biocompatibility [69], to control the corrosion rate [70], and to reduce gaseous hydrogen production [71]. It should be possible to compare magnesium-based materials based on the results of various studies by developing a set of standardized protocols to assess corrosion, biocompatibility, and bone healing. In addition, wider collaboration with clinicians is encouraged to enable the design and development of magnesium-based materials at the earliest stages for specific clinical indications. Continuous efforts and collaboration between scientists and clinicians in different fields should be made to develop excellent and biocompatible fixation materials.

## 5. Conclusions

The stable maintenance of bone fragments through resorbable metallic fixations weakens with time after osteosynthesis. After two months, it is as weak as a polymeric screw made of poly(lactic-coglycolic) acid. Fortunately for stability, the modulus of elasticity of the magnesium alloy is much higher than that of PLGA.

The decrease in the pull-out force that occurs with the progression of screw resorption may cause early bone healing to be vulnerable to displacement. For this reason, it seems clinically reasonable to recommend using more screws and/or combining thinner screws with thicker (more resistant) screws. This approach will provide more stable resorbable metallic osteosynthesis.

## Figures and Tables

**Figure 1 materials-14-00237-f001:**
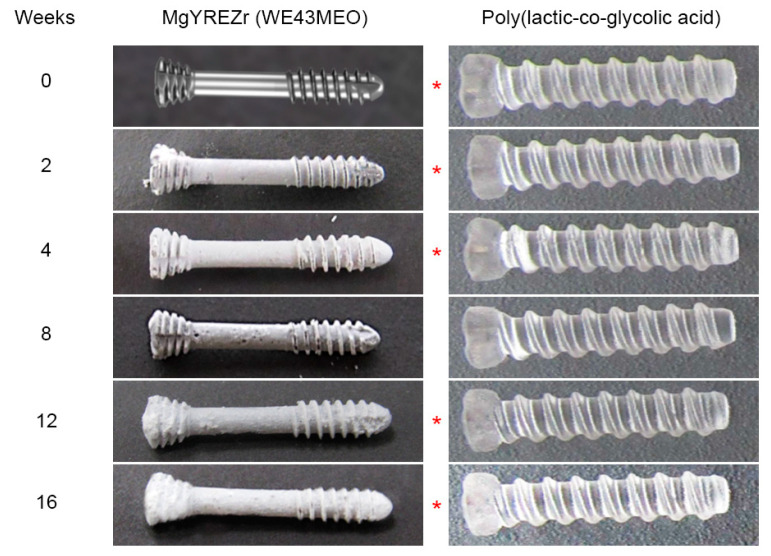
Appearance of samples used in the resorption experiment: magnesium alloy screws on the left-hand and poly(lactic-co-glycolic acid) screw on the right-hand side. All screws had a length of 14 mm. Their length did not change during the experiment. The photographs were not scaled; hence the different lengths in the picture. Asterisks indicate statistical significance in the pull-out force.

**Figure 2 materials-14-00237-f002:**
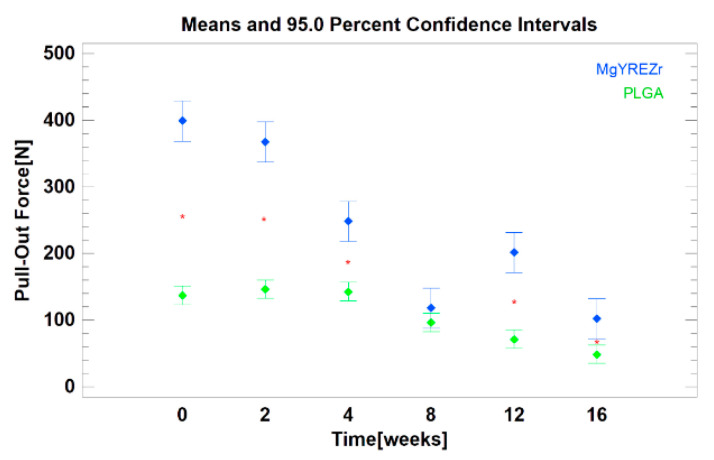
Comparison of axial pull-out force of magnesium alloy screw (blue) versus control: poly(lactic-co-glycolic acid) screw (green). The confidence intervals (brackets), mean values (points), and the statistical difference between two groups for each time point (asterisks) were determined.

**Figure 3 materials-14-00237-f003:**
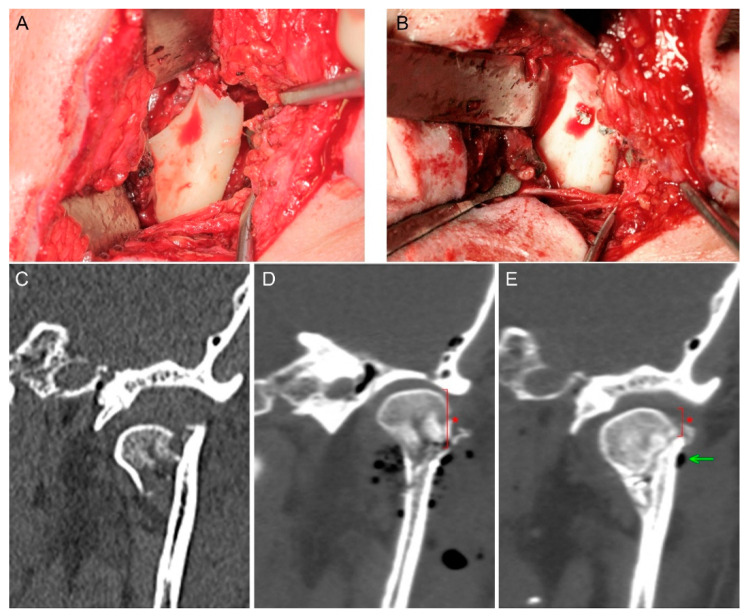
Mandibular head fracture fixation. The phenomenon of reduced mandibular ramus height appeared (asterisk). (**A**) Fracture site. (**B**) Osteosynthesis by two magnesium 1.7 mm × 14 mm screws. (**C**) Pre-op scan—the mandibular head is dislocated downward which shortens the mandibular ramus. (**D**) Fixation by compressive headless screws; visible gas bubbles are the air introduced into the wound during open reduction; height of mandible head is marked by an asterisk. (**E**) 6-month post-op follow-up—fixed bone remodeling, remnants of the produced hydrogen gas (arrow), shortening of the mandibular ramus as a result of the proximal fragment (mandible head) down-shifting along the fissure of the fracture (asterisk).

**Table 1 materials-14-00237-t001:** Axial pull-out force change during screw material resorption (mean ± standard deviation).

Time [Weeks]	MgYREZr [N]	PLGA [N]	Note
0	399 ± 7.5	138 ± 26.5	*p* < 0.01
2	367 ± 28.6	147 ± 4.3	*p* < 0.01
4	249 ± 34.2	143 ± 11.0	*p* < 0.01
8	118 ± 71.1	97 ± 17.3	NS
12	201 ± 27.1	72 ± 27.2	*p* < 0.01
16	102 ± 36.4	49 ± 7.0	*p* < 0.01

MgYREZr—magnesium alloy screw. PLGA—poly(lactic-co-glycolic acid) screw. NS—no statistical significance.

## Data Availability

The data presented in this study are available on request from the corresponding author.

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
