# Peer review of "Change in Pull-Out Force during Resorption of Magnesium Compression Screws for Osteosynthesis of Mandibular Condylar Fractures"

_materials, 2021, doi:10.3390/ma14020237_

Round 1
Reviewer 1 Report
- Long phrases with unclear message (see rows 111-113, 171-177, 151-154, 124, 125 and others)
- English language should be revised (see rows 13, 119 and others)
- There are no measurement units presented in Table 1
- Figure 2 should be presented in the Results section
- The Results section doesn't provide sufficient information
- The text doesn't explain why the Pull-out force at week 8 for MgYREZr is lower than the Pull-out force at 12, respectively 16 weeks for the same alloy.
Reviewer 2 Report
The research has not been well justified and the topic has not been well explored in this manuscript.
Why the pull-out force is required to be measured for biodegradable materials? What is the rationale in terms of clinical application? The significance of using magnesium alloys is clear but the need for the current study must be better justified.
What is the molecular weight of PLGA? PLGA degradation strongly depends on Mw.
Figure 2 needs scale bar to compare the dimensions. The shape and threads for Mg and PLGA screws are different. How do you consider the difference in pull out force caused by the shape discrepancy?
Figure 3 needs error bars and the number of measurements. Also the statistical difference between two groups for each time point should be displayed in the graph.
Why pull out force for MG increases after 8 weeks? In the increase significant?
Reviewer 3 Report
The work is very interesting, it presents a novel result, interesting for readers and of scientific soundness. However, the following comments are presented that I recommend and consider it necessary to be able to publish the manuscript:
- Avoid introducing figures in the introductory section, only commenting on the results of the same and citing it appropriately.
- Extend the last paragraph of the introduction further, commenting on the development of the article presented below.
- Briefly explain the content/conclusion of the article cited in 2020 (corresponds to reference number 30), explaining its results that motivate the performance of the work presented in this manuscript (first paragraph of the "Materials and Methods" section).
- Figure 2 Footer: "righthand side" change to "right-hand side".
- Complete review of the article: expressions like "A p-value of less than ..." (line 89). Line 116: "May be ...".
- The article is out of balance in its structure. It presents a brief introduction in relation to the discussion section, which is too long. I recommend incorporating some paragraphs of the discussion into the introduction, since some paragraphs, especially the first ones, are typical of the introduction section.
- There is no reference in the manuscript to aspects such as inflammation, thermal hypersensitivity, cell apoptosis, oxidative and nitrosative stress in patients treated with magnesium. It is recommended to comment on the aspects found in the patients in the manuscript since they are considered of great importance.
- The conclusion is very brief. It is recommended to extend it to at least twice the length, giving more details of the conclusion of the work presented.
Round 2
Reviewer 3 Report
I recommend publishing the manuscript in its current form.
Author Response
Dear Sir,
Suggested corrections have been introduced in references [red mark].
Best regards,
Author